# SH3Ps—Evolution and Diversity of a Family of Proteins Engaged in Plant Cytokinesis

**DOI:** 10.3390/ijms20225623

**Published:** 2019-11-11

**Authors:** Anežka Baquero Forero, Fatima Cvrčková

**Affiliations:** Department of Experimental Plant Biology, Faculty of Sciences, Charles University, Viničná 5, CZ 128 43 Prague, Czech Republic; anezka.houskova@natur.cuni.cz

**Keywords:** cytokinesis, evolution, phylogeny, cell plate, formin, interaction specificity

## Abstract

SH3P2 (At4g34660), an *Arabidopsis thaliana* SH3 and Bin/amphiphysin/Rvs (BAR) domain-containing protein, was reported to have a specific role in cell plate assembly, unlike its paralogs SH3P1 (At1g31440) and SH3P3 (At4g18060). SH3P family members were also predicted to interact with formins—evolutionarily conserved actin nucleators that participate in microtubule organization and in membrane–cytoskeleton interactions. To trace the origin of functional specialization of plant SH3Ps, we performed phylogenetic analysis of SH3P sequences from selected plant lineages. SH3Ps are present in charophytes, liverworts, mosses, lycophytes, gymnosperms, and angiosperms, but not in volvocal algae, suggesting association of these proteins with phragmoplast-, but not phycoplast-based cell division. Separation of three SH3P clades, represented by SH3P1, SH3P2, and SH3P3 of *A. thaliana*, appears to be a seed plant synapomorphy. In the yeast two hybrid system, Arabidopsis SH3P3, but not SH3P2, binds the FH1 and FH2 domains of the formin FH5 (At5g54650), known to participate in cytokinesis, while an opposite binding specificity was found for the dynamin homolog DRP1A (At5g42080), confirming earlier findings. This suggests that the cytokinetic role of SH3P2 is not due to its interaction with FH5. Possible determinants of interaction specificity of SH3P2 and SH3P3 were identified bioinformatically.

## 1. Introduction

In land plants, cytokinesis is a result of a precisely orchestrated sequence of events that include rearrangements of actin and microtubule cytoskeletons, leading to post-anaphase assembly of the microtubular scaffolding of the phragmoplast at a position that has been previously marked by an “actin-depleted zone”, i.e., an area largely devoid of microfilaments, whose location, in turn, reflects that of the microtubular preprophase band [1,2,3]. Guided by microtubules and aided by dynamins, Golgi-derived membrane vesicles fuse to generate first a tubular meshwork, and then a flat, fenestrated sheet that later develops into a cell plate, which expands simultaneously with the phragmoplast (e.g., [4,5,6]), and finally fuses with the plasmalemma, thereby completing cell division. Besides land plants, also streptophyte algae perform a variant of phragmoplast-based cytokinesis [7], while chlorophyte algae such as *Chlamydomonas* divide their cells by cleavage, employing a different microtubule-based structure termed the phycoplast [8]. Dozens of structures and proteins participating in the intricate process of phragmoplast-based cell division have been already described [9], but the inventory of the complex plant cytokinetic apparatus is still far from complete.

Recently, Arabidopis SH3P2 has been identified as a novel component of the molecular machinery of plant cell division [10]. SH3P2 is one of three Arabidopsis SH3P proteins, characterized by the presence of an N-terminal-conserved Bin/amphiphysin/Rvs (BAR) domain, which is a shorter, plant-specific version of the F-BAR (or FCH-BAR, or EFC for extended FCH homology) domain, joined by a coiled coil segment with a C-terminal-conserved SH3 domain. BAR domains can form crescent-shaped dimers that bind preferentially to highly curved membranes [11] and participate in actin recruitment to curved membranes during clathrin-mediated endocytosis [12]. The second shared domain, SH3 (Sarc homology 3), is a conserved, non-catalytic domain approximately 60 amino acids long, folding into compact beta-barrel consisting of five anti-parallel beta-strands [13]. It is found in proteins of signaling pathways regulating the cytoskeleton, modulating membrane dynamics, or serving as adaptors that link metazoan tyrosine kinases to specific target proteins (see [14]). Opisthokont proteins harboring F-BAR (or BAR) and SH3 domains engage in sensing or modulating membrane curvature and in co-ordinating actin cytoskeleton and membrane remodeling, in part through their interaction with dynamins and with components of the actin nucleating machinery (see [15])—both formins [16,17,18] and regulators of the Arp2/3 complex [15,19]. By analogy with the observations from opisthokonts, SH3 domains of plant SH3Ps were predicted to interact with formins as well [20].

The *Arabidopsis thaliana* genome encodes three members of the SH3P family—SH3P1 (AtSH3P1, At1g31440), SH3P2 (AtSH3P2, At4g34660), and SH3P3 (AtSH3P3, At4g18060). Functional data on all of them are rather fragmentary, but all three proteins appear to be involved in endocytosis. SH3P1 associates with the plasmalemma and endomembranes, co-localizes with clathrin, and is implicated in trafficking of clathrin-coated vesicles and possibly also in vesicle transport along the actin cytoskeleton [21]. The majority of the cellular SH3P2 pool is also associated with clathrin-coated vesicles and late endosomes [22]. SH3P3, but not SH3P1 or SH3P2, interacted in a yeast two hybrid system with ADL6, a dynamin homolog implicated in clathrin-mediated endocytosis, suggesting a possible endocytotic role for this protein as well [23].

Besides its association with clathrin-coated vesicles, SH3P2 localizes to autophagosomes, participates in the membrane deformation process during autophagosome formation, and interacts with the autophagy-related protein ATG8 [24,25]. Inducible RNAi knockdown of SH3P2 suppresses formation of autophagosomes and brings about root growth arrest, premature cotyledon senescence, and subsequent seedling lethality [24]. Both SH3P2 and SH3P3 bind FYVE1, a FYVE domain protein participating in autophagy, degradation of ubiquitinated membrane proteins, and vacuole biogenesis [22]. SH3P2 also interacts with FREE1, another FYVE domain protein that connects the ESCRT-I-dependent endosomal sorting and autophagy pathways [26]. The role of SH3P2 in endosomal sorting and degradation of ubiquitinated integral membrane proteins can be explained by its ability to bind, among other partners, ESCRT-I, ubiquitin [27], and polyubiquitin [28].

Unlike the endocytosis- and autophagy-related functions, which involve more than one member of the SH3P protein family, the cytokinetic role of SH3P2 is specific to this SH3P paralog. Root growth defect of plants with RNAi-silenced SH3P2 turned out to be associated with frequent failure of cytokinesis. This defect cannot be compensated for by the presence of normal levels of SH3P1 or SH3P3 [10], albeit according to publicly available transcriptome data [29], all three genes are expressed throughout vegetative tissues, including the root tip. Thus, SH3P1 and SH3P3 are not functionally redundant with SH3P2 with respect to the cytokinetic function. During cytokinesis, SH3P2 resides at the nascent cell plate and later decorates its broad expanding margin, specifically localizing to the areas exhibiting increased membrane curvature. The protein can form heterodimers with SH3P1 and higher-order complexes involving the dynamins DRP1 and DRP2, which are known to be engaged in cytokinesis. Consequently, both SH3P2 and SH3P1, but not SH3P3, co-immunoprecipitated with DRP1A (At5g42080) in tissue extracts [10].

Here, we attempt to reconstruct the phylogeny of the plant SH3P proteins, examine the predicted interaction between Arabidopsis SH3Ps with a representative of the extensive Arabidopsis formin family [30]—the Class I formin FH5 (AtFH5, At5g54650) previously reported to have a role in cytokinesis [31]—and identify candidate determinants of interaction specificity among angiosperm SH3P paralogs. Our results contribute to characterization of the functional diversity of Arabidopsis SH3Ps and to understanding the origins of the “cytokinetic” SH3P2 type of plant SH3Ps.

## 2. Results

### 2.1. Inventory of SH3P-Encoding Genes in Selected Plant Lineages

We screened sequence data from a total of 20 green plant genomes representing chlorophyte and charophyte algae, liverworts, mosses, lycophytes, gymnosperms, and a variety of angiosperm lineages. The two chlorophytes examined (*Chlamydomonas reinhardtii* and *Volvox carteri*) did not yield any significant homologs of Arabidopsis SH3P1, SH3P2, or SH3P3 (best BLAST hits had the E-value of 0.01). We also did not find any significant homologs in the Rhodophyta and Glaucophyta sections of the GenBank database, or in the predicted proteins of the charophyte *Klebsormidium flaccidum,* whose genome sequence has been published [32]. However, all other genomes examined encoded between one and eight convincing members of the SH3P family, and all seed plants examined possessed at least three SH3P-encoding genes (Table 1). For many of these genes, several predicted alternative transcripts have been found. The full list of identified proteins, including sequence accession numbers and information on alternative transcripts, is provided in Appendix A.

### 2.2. Phylogeny of Plant SH3P Proteins

Protein sequences derived from all found genes (except for two short gymnosperm fragments) were used to reconstruct a phylogenetic tree of the SH3P family. For genes with alternative transcript predictions, a splicing variant whose domain layout was closest to the canonical SH3P protein layout was included in the analysis; as a rule, this was the longest predicted isoform. In the resulting tree (Figure 1), each of the proteins could be unequivocally assigned to one of three well-supported clades, each of them harboring one of the three Arabidopsis sequences. We shall therefore term these three clades SH3P1, SH3P2, and SH3P3 according to their Arabidopsis members. Alternatively spliced transcripts were found in all three clades (see Appendix A).

Remarkably, all the charophyte, liverwort, moss, and lycophyte sequences clustered reliably into the SH3P1 clade. Furthermore, each seed plant species included in our study harbored at least one representative of each of the three clades, except *Pinus taeda,* where the sequence data were obviously incomplete (see Table 1).Together, this suggests that plant SH3Ps evolved from a SH3P1-like ancestor and that the SH3P2 and SH3P3 clades diverged early in seed plant evolution.

Within each clade, evidence for sporadic gene duplication in at least some lineages was observed. There are two SH3P2 clades in the three grasses that were included as representatives of the monocot lineage, suggesting an ancestral gene duplication that has been retained in grass genomes. Other gene duplication events occurred in grapevine and *Solanum* sp. SH3P2s, as well as in the poplar genes of all three clades. In the case of SH3P2, only limited resolution of the deep phylogenetic relationships was achieved even by an analysis that only involved SH3P2 sequences (and therefore allowed inclusion of a greater portion of the protein sequence in non-gapped parts of the alignment). However, we could confirm that the duplication events in dicots were obviously independent from the duplication in grasses, and most likely also from each other, with one duplication event taking place in a common ancestor of the two *Solanum* species examined (Appendix A). 

### 2.3. Differential Binding of Arabidopsis SH3P2 and SH3P3 to the Cytokinetic Formin AtFH5 and to DRP1A

Since SH3P proteins were predicted as putative binding partners of plant formins [20], we employed the yeast two hybrid assay to test the ability of the two derived SH3P paralogs, SH3P2 and SH3P3, to bind the *A. thaliana* Class I formin FH5 that has been previously found to participate in cytokinesis [31]. Since FH5 is a transmembrane protein, only the C-terminal portion of its cytoplasmic part comprising the FH1 and FH2 domains, predicted to participate in possible SH3P interaction, was cloned into the DNA binding domain (DBD) fusion vector pGBKT7, while complete coding sequences of all three SH3Ps were fused with the activation domain (AD) in the pGADT7 vector. Unfortunately, we could not achieve stable yeast co-transformation of SH3P1 and FH5-derived constructs; thus, only SH3P2 and SH3P3 could be evaluated. For control, we also generated a DBD fusion construct derived from DRP1A, a dynamin homolog previously shown to associate with SH3P2, but not SH3P3, by protein co-immunoprecipitation [10]. The results indicate that SH3P3, but not SH3P2, can interact with the C-terminal part of FH5 in our experimental setup, while an opposite specificity was found for DRP1A. This confirms both the previously predicted SH3P–formin interaction and the earlier biochemically detected selective SH3P–DRP1A association [10], documenting thus functional diversification of plant SH3s proteins. An additional control construct involving a DBD fusion of SH3P2 confirmed SH3P2 dimerization [10,11], further documenting functionality of our SH3P2-based activation domain construct (Figure 2).

Remarkably, the yeast transformants harboring the AD fusion of SH3P2, together with the DBD fusion of DRP1, grew poorly under stringent selection conditions, and even under less stringent selection, their growth was noticeably reduced compared to transformants carrying the AD fusion of SH3P2 together with the DBD fusion of the same protein, or those containing the AD fusion of SH3P3 together with the DBD fusion derivative of FH5 (Figure 2). These differences apparently reflect varying interaction strength rather than, e.g., construct toxicity, since all yeast transformants examined grew comparably on -Leu -Trp media that select only for the presence of the two plasmids, but not for two hybrid interactions.While the association between SH3P2 and DRP1A was previously detected biochemically, and confirmed also by in vivo co-localization in stably transformed *A. thaliana* plants [10], the observed interaction between SH3P3 and FH5 might have been an artifact of the yeast two hybrid system. To confirm its possible biological relevance, we generated fluorescent protein fusion derivatives of both interaction partners and examined their ability to co-localize in planta upon heterologous transient expression in the epidermis of Agrobacterium-infiltrated mature *Nicotiana benthamiana* leaves, driven by the strong 35S promoter. Both proteins largely, though not completely, co-localized in a patchy pattern in the cytoplasm of epidermal pavement cells (Figure 3). While the observed localization may reflect artifactual protein aggregation, due to an unnaturally high expression level, it is nevertheless consistent with both proteins being capable of in vivo interaction.

### 2.4. Possible Determinants of Interaction Specificity Among Angiosperm SH3P Proteins

To gain insight into the mechanistic aspects of the observed and previously reported SH3P interaction specificity, we took advantage of the good sequence conservation among angiosperm SH3Ps that enabled us to visually identify positions where amino acid type [33] is shared across all clades or specific to a particular clade (Figure 4, Appendix A). 

In total, 14 candidate clade-specific conserved sites were found, most of them within the BAR domain (Table 2, Figure 4). Remarkably, only two clade-specific conserved positions were identified for the SH3P1 clade, while SH3P2 proteins shared 4 such positions and the SH3P3 clade shared 8 specific sites.

To further explore the possible role of the clade-specific conserved positions in determining the proteins’ interaction specificity, we constructed three-dimensional (3D) models of all three *A. thaliana* SH3P proteins. Initial attempts using the Phyre2 [34] or Swiss-Model [35] software generated only partial models of either the BAR or SH3 domains. However, models containing both domains have been successfully generated by the RaptorX algorithm [36]. While 100% of the SH3P2 and SH3P3 sequences could be modeled using this approach, in the case of SH3P1, the model reliably covered only 74% of the sequence, with unreliable prediction for the long disordered linker between the two conserved domains (Appendix A). It is worth noting that in both *A. thaliana* and *A. lyrata,* this linker contains an insertion that may be specific to *Arabidopis,* or perhaps to Brassicaceae, since it is absent in other SH3P sequences examined in our study (for angiosperm sequence alignment, see Appendix A).

Position of conserved sites was subsequently mapped on the models. While all the clade-specific conserved amino acids were predicted to exhibit at least some surface exposure, sites within the BAR domain were often semi-buried or located at the predicted membrane interaction or BAR domain dimerization interface, and only those located in or close to the SH3 domain of SH3P2 and SH3P3 were exposed at prominent surface positions (Figure 5, Table 2).Those sites, three in SH3P2 and two in SH3P3, are thus the most likely to contribute to the choice of interacting partners.

## 3. Discussion

In a recent study, Arabidopsis SH3P2, a member of the family of plant SH3Ps (BAR and SH3 domain-containing proteins), was identified as an important player in the cytokinesis machinery, required for proper cell plate formation, localizing to the nascent cell plate and its expanding margins, and engaging in molecular interactions with dynamin-related proteins DRP1 and DRP2 [10]. Its cell plate localization has since been confirmed in another report [37]. 

The plant SH3Ps are members of a broader category of proteins containing two evolutionarily conserved domains—the membrane-binding BAR (Bin/amphiphysin/Rvs) or F-BAR domain [11] and the SH3 (Sarc homology 3) domain mediating protein–protein interactions [14,38]. Such proteins are found also in opisthokonts, where they participate in connecting membrane dynamics, signaling, and the actin cytoskeleton (see [15]). The BAR domain dimer can bind to membrane regions exhibiting increased curvature or enriched in membrane curvature-promoting phospholipids [39], enabling the proteins to sense or modulate membrane curvature. They also interact with a plethora of signaling and regulatory proteins through their SH3 domains [38]. Opisthokont BAR–SH3 proteins, such as amphiphysin, endophilin, or syndapin, have been documented to interact with dynamins, especially during clathrin-mediated endocytosis [40,41]. The ability of plant SH3Ps to bind dynamins or dynamin-related proteins [10,23] and their role in clathrin-mediated endocytosis [23] is thus not surprising.

In spite of dramatic differences in the mechanics of cytokinesis among opisthokonts and land plants (see, e.g., [42]), engagement of BAR- and SH3 domain-containing proteins in cytokinesis is not a feature unique to plants. In the fission yeast *Schizosaccharomyces ponbe*, F-BAR- and SH3 domain- containing proteins Cdc15 and Imp2 provide a scaffold for contractile ring assembly at the division site, which involves a highly curved membrane furrow [43,44]. A similar mechanism operates also in the budding yeast [45] and in metazoan cells [46].

Could, thus, the newly discovered role of SH3P2 in Arabidopsis cytokinesis reflect an ancient cellular mechanism, perhaps inherited from the last eukaryotic common ancestor? The apparent absence of SH3P-encoding sequences in glaucophyte, rhodophyte, and chlorophyte genomes, together with their consistent presence (at least) in liverworts, mosses, and vascular plants, makes such a scenario rather unlikely, since it would require independent losses of the gene family, as well as evolution of alternative solutions for cytokinesis in the glaucophytes, red and green algae. Instead, we propose that the plant SH3Ps arose by new combination of ancient domains, not earlier than in the ancestors of streptophytes. We hypothesize that the origin of plant SH3Ps correlates with the appearance of a phragmoplast-, rather than phycoplast-based mechanism of cell division [7,8]. Nevertheless, phycoplast-mediated cytokinesis also involves orchestrated interplay of microtubule and membrane rearrangements with involvement of dynamins [47], similar to phragmoplast-based division. SH3Ps cannot thus obviously be essential for the process of cell division per se.

To our surprise, we obtained conflicting results when searching for SH3P homologs in charophyte genomes. We found no significant homologs in available sequence data from *Klebsormidium flaccidum*, but identified one member of the SH3P gene family in the related species *K. nitens*. This is most likely an artefact due to incomplete coverage of the published *K. flaccidum* genome sequence, which may cover as little as 75% of its nuclear genome [32]. Since two SH3P homologs were identified in *Chara braunii* in our bioinformatic searches, and a SH3P family member was also experimentally found to be differentially represented in the membrane-associated proteome of acidic and alkaline cell domains in the related *C. australis* [48], we are convinced that charophytes do possess SH3P proteins, similar to land plants.

Our reconstruction of plant SH3P phylogeny suggests that an ancestral streptophyte SH3P family, corresponding to the extant angiosperm SH3P1 clade, gave rise to two derived clades, SH3P2 and SH3P3, in early seed plants. This model agrees well with a previous, mostly dicot-oriented phylogeny based on a narrower species sampling [27]. During subsequent evolution, members of all clades underwent sporadic gene duplications in angiosperms. Most of these duplications took place in the cytokinesis-associated SH2P2 clade, where duplicated genes were detected in grasses, Solanaceae, and the grapevine. However, some species, including *A. thaliana*, can obviously live (and divide their cells) with a single SH3P2 copy. 

The three clades of angiosperm SH3Ps have clearly diverged functionally, as demonstrated not only by the unique role of SH3P2 in cytokinesis [10]. There are also documented inter-paralog differences in the binding to dynamins and dynamin-related proteins [10,23] or components of the autophagy, protein degradation, or endocytotic machineries ([22,26,27], see also Introduction). In this study, we documented differences in the ability of SH3P2 and SH3P3 to bind the cytokinesis-associated formin FH5—a member of a conserved actin nucleator family, predicted as a possible interaction partner of BAR–SH3 proteins [20], while confirming the previously described differences in binding specificity of SH3P2 and SH3P3 towards the dynamin homolog DRP1A, inferred from immunoprecipitation experiments and in vivo protein co-localization [10]. Unfortunately, we were so far unable to examine possible binding between SH3P1 and FH5 because we repeatedly could not achieve stable co-transformation of the corresponding two hybrid constructs in yeast, even under conditions where combinations of single constructs and empty partner vectors transformed well. This may suggest an interaction whose product is deleterious for the yeast host. However, additional experiments, involving at least use of an alternative set of vectors, would be required to confirm this suspicion. Rather than examining the causes of the co-transformation failure, we focused on confirming the newly found interaction between SH3P3 and FH5 by independent means.

Consistent with the SH3P3—FH5 interaction being biologically relevant, fluorescent protein—tagged derivatives of these two proteins co-localized in a transient expression assay in planta. Although the observed structures may possibly result from protein aggregation due to heterologous overexpression, they nevertheless show that the two proteins can meet, and possibly bind each other, in vivo. The list of possible SH3P binding partners, however, does not end with proteins engaged in the control of membrane trafficking or cytoskeleton–membrane co-ordination. Similar to their opisthokont relatives [15], plant SH3 proteins participate in additional regulatory and signaling processes. A truncated version of SH3P1 was recovered in modified yeast two hybrid screen for interactors of the jasmonic acid signaling protein NINJA [49], and SH3P3 interacts with the BRL3 signalosome component [50]. Selective interaction with these or other signaling or regulatory proteins may present another opportunity for functional diversification among SH3P paralogs.

We took advantage of the collection of SH3P sequences assembled for the purpose of our phylogenetic study and used these data to identify possible clade-specific determinants of interaction specificity. We found several such candidate sites, mostly clustered at or close to the SH3 domain that is considered a major protein–protein interaction interface [38], in both SH3P2 and SH3P3, but not in SH3P1. This is consistent with the hypothesis that SH3P1 corresponds to an ancestral, presumably non-specialized state, while SH3P2 and SH3P3 are derived (and more specialized) paralogs. However, biological significance of the proposed protein–protein interaction specificity determinants will need to be tested experimentally.

In summary, we provide bioinformatic evidence that SH3P proteins are a streptophyte synapomorphy and that the three SH3P clades of extant angiosperms diversified already in the common ancestor of seed plants. We also show that SH3P2 and SH3P3 differ in their ability to bind the cytokinesis-associated formin FH5 and the dynamin-related protein DRP1A, suggesting that the cytokinetic role of SH3P2, previously proposed to engage DRP1 [10], is not due to its interaction with FH5. Last but not least, based on analysis of a representative angiosperm SH3P sequence collection, we propose candidate interaction specificity determinants that may be responsible for the different spectra of binding partners of individual SH3P paralogs.

## 4. Materials and Methods 

To identify plant SH3P protein sequences, publicly available databases were searched using the BLAST algorithm [51] with the three known *A. thaliana* SH3P sequences, retrieved from Uniprot based on their annotation, as queries. Plant genome-specific databases, in particular Phytozome ([52]; for multiple species, as listed in Appendix A) or Congenie ([53]; for gymnosperms), were preferred. For species not covered by these databases, as well as for identification of additional homologs, GenBank searches were performed with the narrowest taxonomical restriction possible. Only hits with E better than e-30 were considered as positives.

Alignment of protein sequences for the phylogenetic study and for identifying conserved sites was performed by a combination of algorithmic and manual methods, as described previously [54]. Phylogenetic trees were constructed from these data after exclusion of gap-containing positions using the maximum likelihood algorithm implemented in the MEGA X software package [55] with default parameters, except for using 500 bootstrap replicates.

To clone the SH3P2 and SH3P3 cDNAs, 5-day-old *A. thaliana* seedlings grown in vitro, as described previously [56], were used to isolate RNA with an RNeasy Plant Mini Kit (Quiagen). Obtained material served as a template for cDNA production by a Transcriptor High Fidelity cDNA Synthesis Kit (Roche), according to manufacturer’s instructions, and resulting cDNA was used as a template for PCR amplification of the fragments of interest using primers listed in Appendix A. For cloning of the FH1 + FH2 domains of FH5 (residues 263–896), a cDNA clone already available in our laboratory [57] served as a template. PCR was performed using the primers listed in Appendix A, and its products cloned by the restriction–ligation method using the indicated restriction enzymes into the pGADT7 and pGBKT7 yeast two hybrid vectors (Clontech/Takara, Cat. No. 630442 and 630443). Additional constructs were generated in Gateway® compatible, modified vectors [58] using PCR with primers listed in Appendix A. DRP1A coding sequence was re-cloned into the Gateway® donor vector pDONOR221 from a published cDNA [59] by amplification using primers with recombination sites followed by BP reaction (BP clonase, Invitrogen). SH3P2 was re-cloned into pDONOR221 from previously created pGAD vector by the same method. To clone SH3P1, a cDNA clone from the RIKEN Arabidopsis full-length clone collection (resource number pda07807) was used as a template to generate entry vector SH3P1_pDONOR221 by BP reaction from the PCR product, as described above. Subsequently, PCR amplifications of the coding sequences with added terminal recombination sites were performed and the PCR products were recombined with the vector pGBKT-gw by standard LR reaction with LR clonase II, according to manufactures instructions (Invitrogen). Integrity of all fusion constructs was verified by sequencing prior to yeast transformation.

For the yeast two hybrid assay, the MATCHMAKER GAL4 Two-Hybrid System 3 (Clontech) was employed, according to manufacturers´ instructions. The yeast strain AH109 (*MATa, trp1-901, leu2-3, 112, ura3-52, his3-200, gal4Δ, gal80Δ, LYS2::GAL1UAS-GAL1TATA-HIS3, GAL2UAS-GAL2TATAADE2, URA3::MEL1UAS-MEL1TATA-lacZ)* was co-transformed according to the manufacturer’s manual, with vectors containing fusion of GAL4 DNA binding domain (DBD; vector pGBKT7) or GAL4 activation domain (AD; pGADT7 vector) to create the indicated construct combinations, using empty vectors for control. Transformed yeasts were selected on –Leu –Trp dropout media. To test the protein–protein interactions, transformed yeasts were plated as a dilution series starting with OD_600nm_ = 0.1 either on quadruple dropout medium (-Leu, -Trp, -His, -Ade) for stringent selection, or on triple dropout -Leu, -Trp, -His medium supplemented by 5 mM 3-aminotriazole for less stringent selection.

To prepare the vectors for transient expression experiment, multisite Gateway® building blocks [60] were used. For preparation of 35S:SH3P3:RFP, multisite Gateway® reaction (LR clonase II, Invitrogen) was performed with building blocks: pEN L4-2-R1 (35S promoter), pEN R2-R-L3 (RFP for C-terminal fusion), SH3P3_pDONOR221 (generated as described above), and pB7m34GW. For preparation of 35S:FH5:GFP, first the vector FH5_pDONOR was generated by amplification of coding sequence with recombination sites from available cDNA clone, followed by BP reaction. FH5_pDONOR221 was used for multisite Gateway® reaction, together with vectors pEN L4-2-R1, pEN-R2-F-L3 (GFP for C-terminal fusion), and destination vector pB7m34GW, to build the final vector. The primers used for cloning all Gateway® compatible vectors are listed in Appendix A.

Transient transformation of *N. benthamiana* leaves was performed as described in [61]. Confocal laser scanning microscopy images of leaf epidermis were acquired 2 days post-infiltration, as described previously [62], employing a Zeiss LSM880 microscope with C-Apochromat 40× / 1.2 W objective, 488-nm argon and 561-nm laser for excitation. GFP was detected at 500–525 nm, and RFP at 600–640 nm. Artificial coloring of images and Z-stack montage was performed using ImageJ [63].

For generating 3D molecular models, Arabidopsis SH3P protein sequences were analyzed using the RaptorX software [36] with standard (default) settings. For all three proteins, the resulting model was built from two domains (BAR and SH3), with the BAR domain modeled in all three cases on the PDB 6h7w (fungal retromer-Vps5 complex) chain B template. The SH3 domain of SH3P1 was modeled based on PDB 2cub (SH3 domain of human Nck1), while for both SH3P2 and SH3P3, models of this domain were built on a consensus of PDB templates 2pqh (rat chimeric SH3 domain), 6gbu chain B (fourth SH3 domain of human TSN1), 3reb chain B (SH3 domain of human Hck), 6cq7 (SH3 domain of human MLK3), and 6h5t (human Intersectin SH3A). The overall model quality score (uGDT) was, in all cases, greater than 135 (with values over 50 usually considered as good for sequences over 100 residues), with the normalized GDT score in the range of 33–37 (reflecting the presence of disordered and thus unreliably modeled sequence segments). The *p*-value of the predicted model being worse than the best of a set of randomly-generated models was, in all cases, better than 3. 10^−5^. Model visualizations were generated using the DeepView software [64].

## Figures and Tables

**Figure 1 ijms-20-05623-f001:**
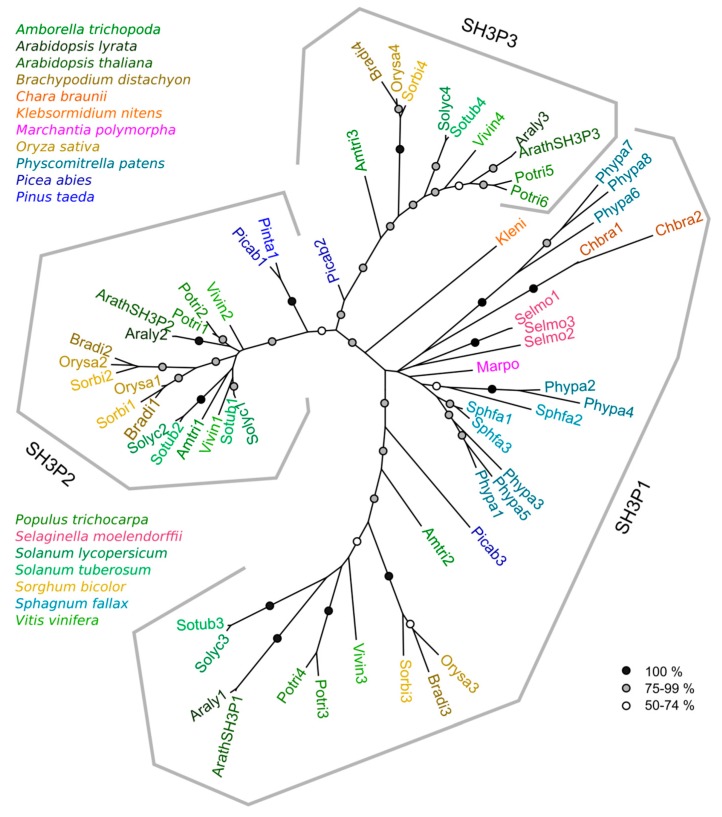
A maximum likelihood phylogenetic tree of plant SH3P protein sequences. Symbols at branches denote bootstrap support (no symbols for support below 50%). For accession numbers of all sequences, see Appendix A.

**Figure 2 ijms-20-05623-f002:**
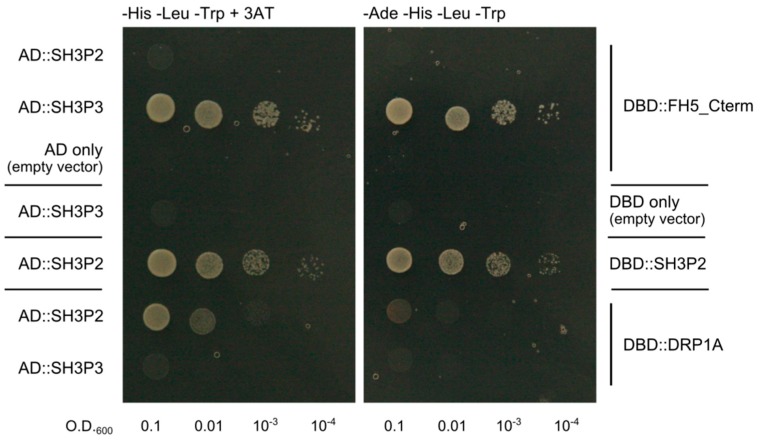
Results of a yeast two hybrid experiment showing that SH3P3, but not SH3P2, can bind the C-terminal cytoplasmic fragment of FH5 comprising the FH1 and FH2 domains (FH5_Cterm), while only SH3P2 but not SH3P3 can interact with the dynamin homolog DRP1A. Dimerization of SH3P2 is also demonstrated. Constructs marked as “DBD” are derived from the pGBKT7 vector and carry the DNA binding domain of the yeast GAL4 transcription factor; those labeled as “AD” are derived from the pGADT7 vector and carry the GAL4 activation domain. Yeast suspensions were plated on two types of substrate—a standard -His -Leu -Trp dropout medium containing 5 mM 3-aminotriazole (3AT; **left**), and a -Ade -His -Leu -Trp dropout medium (**right**) providing more stringent selection. Calculated O.D._600_ of the plated yeast suspension is indicated.

**Figure 3 ijms-20-05623-f003:**
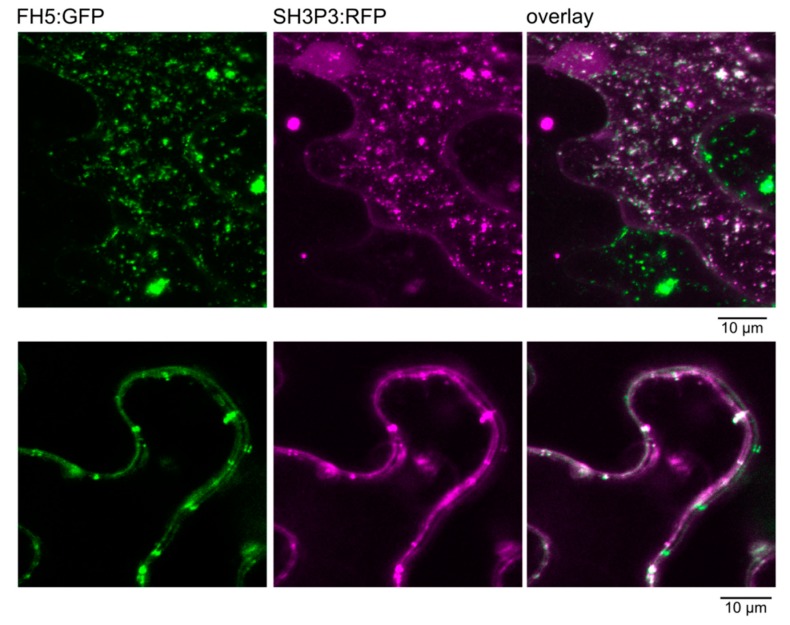
Co-localization of FH5, C-terminally tagged by green fluorescent protein (GFP), with SH3P3, C-terminally tagged by red fluorescent protein (RFP), in transiently transformed *N. benthamiana* leaf epidermis, as documented by confocal laser scanning microscopy. Note the cell exhibiting only GFP fluorescence and the weak nuclear RFP signal, indicating absence of cross-channel signal leak. Top: Maximum intensity projection of a Z-stack of optical sections across an epidermal pavement cell. Bottom: A single confocal section showing a detail of an anticlinal cell to cell boundary.

**Figure 4 ijms-20-05623-f004:**
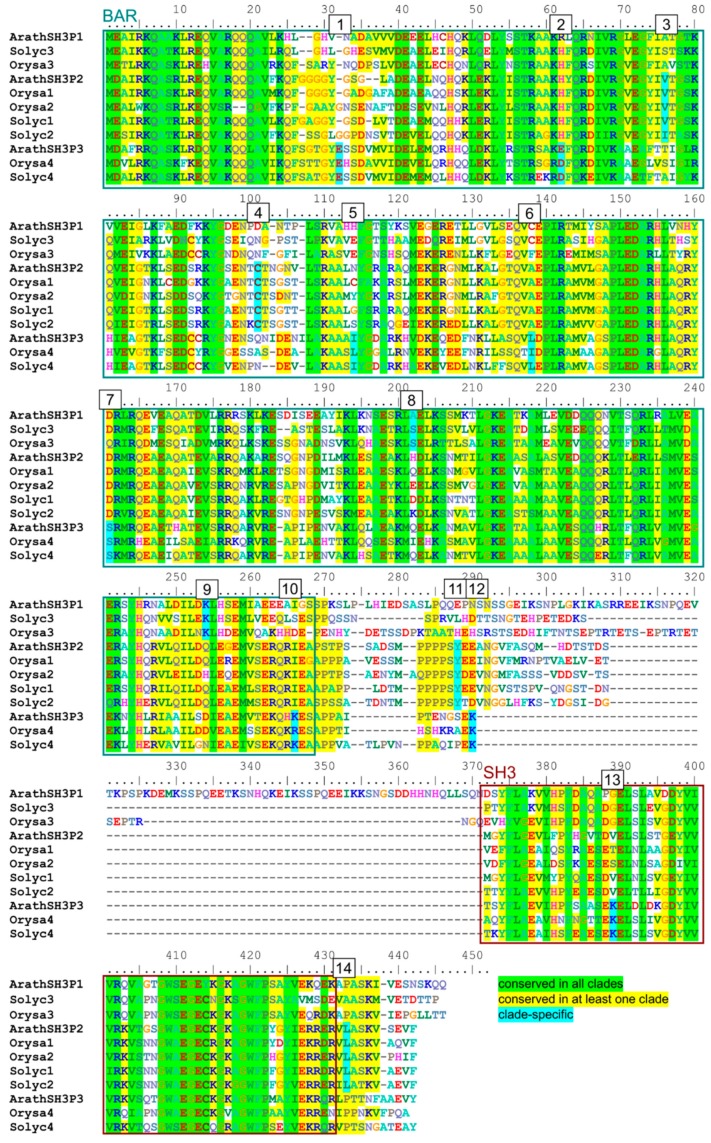
Alignment of representative angiosperm SH3P sequences. Conserved positions, as determined from a larger alignment, including all full-length angiosperm SH3P sequences (see Appendix A), are highlighted in colour. Sites with amino acid category conserved across all sequences are shown in green, those with amino acid category shared by all members of at least one clade and present in at least one member of another clade in yellow, and clade-specific positions (i.e., those where amino acid type is shared by all members of one clade but not with other clades) are highlighted in turquoise and numbered. Domain borders are shown based on annotation of the Uniprot record for Arabidopsis SH3Ps. Amino acid categories are defined as follows: Aliphatic (MILV), small hydrophilic (STPAG), aromatic (FYW), cysteine (C), acid and acid amide (NDEQ), basic (HRK). For sequence descriptions, see Appendix A.

**Figure 5 ijms-20-05623-f005:**
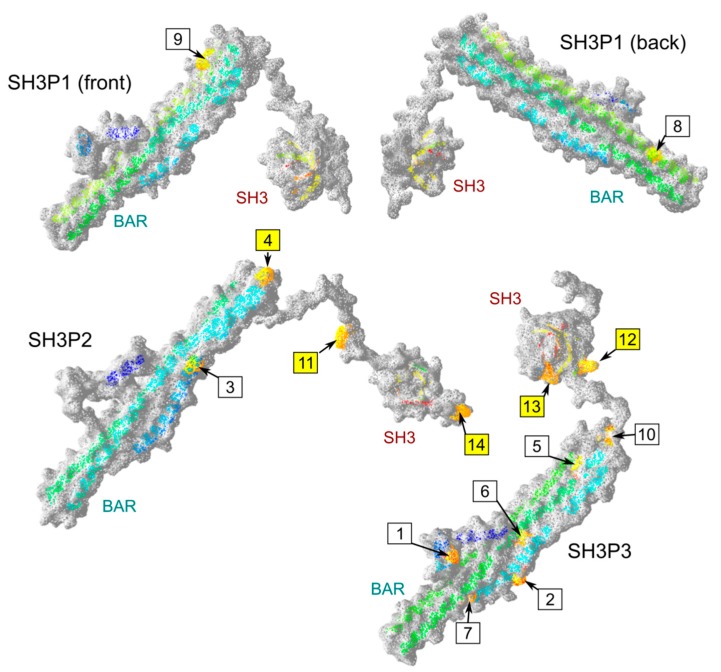
Structural models of Arabidopsis SH3P proteins. Parts of the molecular surface contributed by clade-specific conserved sites (numbered as in Table 2 and Figure 4) are shown in orange, and backbone structure is indicated by a ribbon, colored using the secondary structure succession rainbow scheme (from N-terminus in blue within the BAR domain towards C-terminus in red within the SH3 domain). For SH3P1, two views are presented. Candidate highly exposed interaction sites are marked by labels with yellow background.

**Table 1 ijms-20-05623-t001:** Numbers of identified SH3P paralogs in selected plant species. The inventory may still be incomplete for some, in particular non-model, plants.

Group	Species	Total SH3Ps	SH3P1	SH3P2	SH3P3
Chlorophyta	*Chlamydomonas reinhardtii*	none	-	-	-
	*Volvox carteri*	none	-	-	-
Charophyta	*Chara braunii*	2	2	none	none
	*Klebsormidium nitens*	1	1	none	none
Liverworts	*Marchantia polymorpha*	1	1	none	none
Mosses	*Physcomitrella patens*	8	8	none	none
	*Sphagnum fallax*	5	5	none	none
Lycophyta	*Selaginella moelendorffii*	3	3	none	none
Gymnosperms	*Picea abies*	3	1	1	1
	*Pinus taeda* ^1^	3	?	1	1?
Angiosperms/Dicots	*Amborella trichopoda*	3	1	1	1
Angiosperms/Dicots/Rosidae	*Arabidopsis thaliana*	3	1	1	1
	*Arabidopsis lyrata*	3	1	1	1
	*Populus trichocarpa*	6	2	2	2
	*Vitis vinifera*	4	1	2	1
Angiosperms/Dicots/Asteridae	*Solanum lycopersicum*	4	1	2	1
	*Solanum tuberosum*	4	1	2	1
Angiosperms/Monocots	*Brachypodium distachyon*	4	1	2	1
	*Oryza sativa japonica*	4	1	2	1
	*Sorghum bicolor*	4	1	2	1

^1^ Two fragmentary *P. taeda* sequences could not be reliably assigned to a clade (marked by ?).

**Table 2 ijms-20-05623-t002:** Putative clade-specific conserved sites in angiosperm SH3P protein sequences. For site numbering, see Figure 4, and for source alignment see Appendix A. In case of multiple options for a given site, amino acids are ordered by frequency of occurrence. The predicted extent of surface exposure was estimated visually based on theoretical structure models (see Figure 5 and Appendix A) and expressed in semi-quantitative categories (+/− semi-buried, + partial surface exposure, + + most of sidechain exposed at a prominent part of the molecule surface). AA—amino acid. Standard IUPAC abbreviations are used.

Site	Clade	Domain	Position ^1^	Conserved AA	AA in Other Clades	Exposed
1	SH3P3	BAR	32	E	G or gap	+
2	SH3P3	BAR	62	D or E	H or R	+ ^2^
3	SH3P2	BAR	73	V or I	small hydrophilic (STA)	+/− ^2^
4	SH3P2	BAR	98	C	various	+ +
5	SH3P3	BAR	114	I or L	various	+/− ^2^
6	SH3P3	BAR	138	L or I	various	+/− ^2^
7	SH3P3	BAR	161	S	acid or amide (DEQ)	+/−^2^
8	SH3P1	BAR	197	small hydrophilic (SAT)	various charged	+/−^2^
9	SH3P1	BAR	248	K	various (mostly QDE)	+ ^2^
10	SH3P3	BAR	265	K or R	various (mostly I)	+/−
11	SH3P2	linker	281	Y	various	+ +
12	SH3P3	linker	280	K	various (mostly NDE)	+ +
13	SH3P3	SH3	298	K	various	+ +
14	SH3P2	C-terminal	360	L	small hydrophilic (PAS)	+ +

^1^ Numbering according to the *A. thaliana* representative of the clade where the position is conserved. ^2.^ Located within the three main helices of the BAR domain that engage in membrane binding or BAR domain dimerization [15].

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
