# Peer review of "SH3Ps—Evolution and Diversity of a Family of Proteins Engaged in Plant Cytokinesis"

_ijms, 2019, doi:10.3390/ijms20225623_

Round 1

Reviewer 1 Report

Requested corrections were completed.

Author Response

Thank you very much for favourable evaluation! We have performed an additional spelling check (in addition to some minor additions requested by the other reviewer).

Reviewer 2 Report

This version is improved after the authors added additional data, analysis, and edits.

Interaction of SH3P1 with FH5 was not examined due to the failure of co-transformation of SH3P1 and FH5 constructs though the authors tried a few trials. It is not a positive result but it is worth to give more details how the authors performed the experiments and the possible causes to the failure in the Discussion section. These information could help other potential researchers avoid similar valueless efforts.  

Author Response

Thanks for the suggestions. We have now added a more detailed description of the failed attempts to test SH3P1 interactions (both in Results and Methods, including details of the constructs tested), as well as a paragraph in the Discussion on possible causes of the failure.

This manuscript is a resubmission of an earlier submission. The following is a list of the peer review reports and author responses from that submission.

Round 1

Reviewer 1 Report

SH3P family play important roles in plant cytokinesis. Clarifying the evolution and functional diversity of the SH3P family could contribute knowledge to cell biology. This study can be improved with more covincing experimental data. 

There are only three SH3P members in Arabidopsis. They all probably have important or even unique function. The role SH3P1 in the interaction with FH5 was not determined due to the failure of experiment. It is important to have the result of interaction of SH3P1 with FH5. It is hard to exclude the possibility that the failure of yeast hybrid assay cause the non-interaction of SH3P2 and FH5. SH3P2 has been shown to interact with DRP1. It is better to add the examination of SH3P2 and FH5 interaction as a positive control in this study.

Author Response

Thank you for suggestions of additional experiments, whose results, indeed, provided important new insights.

We now include additional positive control experiments confirming that the observed lack of interaction between SH3P2 and FH5 was not due to non-functionality of our SH3P2-derived construct, since we were able to document SH3P2 dimerization in the two-hybrid assay. We also provide new two-hybrid evidence for the SH3P2-DRP1A interaction that was previously reported based on co-immunoprecipitation and co-localization data. As far as FH5 is concerned, the functionality of our two-hybrid construct is proven by the detected interaction with SH3P3 (including appropriate negative controls). As a further support of the newly reported SH3P3 – FH5 interaction, we are now including additional supportive data from a heterologous tagged protein expression experiment showing that the two proteins can co-localize in planta.

We cannot but agree that yeast two hybrid interaction data for SH3P1 and FH5 would have been a valuable addition. However, we could not achieve stable yeast transformation with the corresponding construct combinations in repeated experiments where single transformations worked well; we believe that this was due to a biological reason (e.g. construct or interaction product toxicity rather than technical failure ). Nevertheless, our conclusions regarding interaction diversity of the two derived SH3P subfamilies (SH3P2 and SH3P3) do not depend on SH3P1 interaction data, and we consider them worth reporting even without experimental data on the third member of the SH3P family.

Reviewer 2 Report

Authors demonstrated that SH3P2, an Arabidopsis thaliana SH3 and BAR domain containing protein, was recently shown to have a specific role in cell plate assembly, not shared by its paralogs SH3P1 and SH3P3. In the yeast two hybrid system, Arabidopsis SH3P3, but not SH3P2, binds the FH1 and FH2 domains of AtFH5, a formin known to participate in cytokinesis, suggesting that the cytokinetic role of SH3P2 is not due to its interaction with AtFH5. Possible determinants of interaction specificity of SH3P2 and SH3P3 were identified bioinformatically. Although the overall interest and visibility of this work, some aspects should still be considered to improve the quality and objectiveness of this work.

Major Comments:

Authors provided only bioinformatics evidence that SH3P proteins. If possible authors need to provide plant studies with molecular conformation. I feel lack of experimental studies in this paper. Author needs to do some additional experiments such as molecular studies or microscopically studies. Abstract is not clear. Abstract needs clear background, objectives, methods, results and conclusion. But the present form of abstract is not clear. Background of the study should be made to very clear. Introduction” is inappropriate and some statements are given without reference citation so rewrite it only within 1 and half page as: First of all present the background studies about the topic in a manner that set a foundation to understand the research problem with proper reference citations. Materials and Methods sections need very clear methods. Author provided only numbers but its not easy to understand readers. Overall, this manuscript written is very poor. English of the MS needs to be critically improved. Conclusion part is not clear.

Author Response

We had a hard time understanding some of the reviewer´s comments, and at places we were unable to decipher what their intended meaning was.

While it may not have been immediately obvious to the Editor, a substantial part of the introductory summarizing paragraph of the review (i.e. all of it besides the first three words and the last sentence) is a collage of two verbatim copied fragments from the abstract of our manuscript, with some (hopefully unintentional) changes of meaning. We really did not claim to demonstrate what others published before! Curiously, in this summary, the reviewer entirely leaves out the central message of our study, whose main focus is the phylogeny of the SH3P family. I also feel somewhat offended by the reviewer´s statement that the objectiveness of our work should be improved (without any rationale provided).

What follows is a point by point reply to reviewer´s major comments (reviewer´s text in purple).

Authors provided only bioinformatics evidence that SH3P proteins.

We do not understand what the reviewer means – the second part of the sentence is obviously missing.

If possible authors need to provide plant studies with molecular conformation. I feel lack of experimental studies in this paper. Author needs to do some additional experiments such as molecular studies or microscopically studies.

We are not sure what is meant by “plant studies with molecular conformation”, but we have the impression that the reviewer does not recognize bioinformatics as a valid method of studying evolution and diversity of proteins. We hope that the Editor does not share the reviewer´s view, which, in our opinion, appears to deviate from the current scientific mainstream. Nevertheless, we are now including additional experimental material (as inspired by comments by the other reviewer), including in vivo protein localization data.

Abstract is not clear. Abstract needs clear background, objectives, methods, results and conclusion. But the present form of abstract is not clear.

Yet, the abstract was good enough for the reviewer to plagiarize as his summary of our study!

Background of the study should be made to very clear. Introduction” is inappropriate and some statements are given without reference citation so rewrite it only within 1 and half page as: First of all present the background studies about the topic in a manner that set a foundation to understand the research problem with proper reference citations.

This section of the review is very vague, and, as we understand it, in part also very obviously counterfactual. How can we shorten an introduction to 1 and 1/2 page, if it is shorter to begin with? Or did we misunderstand the reviewer? We also, even after careful re-reading of the introduction, could not find any statement not supported by references (possibly except very general textbook-level knowledge), so we would have appreciated the reviewer being more specific.

Materials and Methods sections need very clear methods. Author provided only numbers but its not easy to understand readers.

There are, and were, no numbers in the methods, except of numbers of references, parts of plasmid names and information on parameters used in some experimental and bioinformatic analyses. We consider all these “numbers” essential to ensure reproducibility of our methodology. The methods are and were described in the text in a standard manner (we believe that with appropriate level of detail). Thus, this looks like another obviously counter-factual claim by the reviewer.

Overall, this manuscript written is very poor. English of the MS needs to be critically improved. Conclusion part is not clear.

We are afraid that the language problem appears to be rather on the reviewer´s side. In fact, we are surprised that this reviewer feels qualified to judge our English language and style, having very obvious problems with English him-or herself.